# Auto-PINN: Understanding and Optimizing Physics-Informed Neural Architecture

**Yicheng Wang**
Texas A&M University
wangyc@tamu.edu

**Xiaotian Han**
Texas A&M University
han@tamu.edu

**Chia-Yuan Chang**
Texas A&M University
cychang@tamu.edu

**Daochen Zha**
Rice University
daochen.zha@rice.edu

**Ulisses Braga-Neto**
Texas A&M University
ulisses@tamu.edu

**Xia Hu**
Rice University
xia.hu@rice.edu

## Abstract

Physics-Informed Neural Networks (PINNs) are revolutionizing science and engineering practices by harnessing the power of deep learning for scientific computation. The neural architecture's hyperparameters significantly impact the efficiency and accuracy of the PINN solver. However, optimizing these hyperparameters remains an open and challenging problem because of the large search space and the difficulty in identifying a suitable search objective for PDEs. In this paper, we propose Auto-PINN, the first systematic, automated hyperparameter optimization approach for PINNs, which employs Neural Architecture Search (NAS) techniques for PINN design. Auto-PINN avoids manually or exhaustively searching the hyperparameter space associated with PINNs. A comprehensive set of pre-experiments, using standard PDE benchmarks, enables us to probe the structure-performance relationship in PINNs. We discover that the different hyperparameters can be decoupled and that the training loss function of PINNs serves as an effective search objective. Comparison experiments with baseline methods demonstrate that Auto-PINN produces neural architectures with superior stability and accuracy over alternative baselines.

## 1 Introduction

Physics-informed neural networks (PINNs) [20] are promising partial differential equations (PDE) solvers that integrate machine learning with physical laws. Benefiting from the strong expressive power of deep neural networks, PINNs are widely adopted to solve various real-world problems, such as fluid mechanics[6, 9, 23], material science [8, 24, 25] and biomedical engineering [22, 10, 14]. PINNs do not require the time-consuming construction of elaborate grids, and can therefore be applied more easily to irregular and high-dimensional domains than traditional PDE solvers can.

The structures of PINNs are usually simple multilayer perceptrons (MLPs). Similar to other deep learning tasks, the neural architecture configurations of MLP networks, such as depths/widths, and activation functions, have a great effect on the performance of PINNs. However, there is little research on this problem. For instance, Raissi et al. [20] found that increasing the width and depth of PINNs can improve the predictive accuracy, but their experiments are limited to a single PDE problem within a very small search space. While the Tanh activation function is the default option for PINNs, some studies [2, 16] report that Sigmoid or Swish [21] functions are more effective in some cases. However, they did not reach a conclusion about which activation function is preferred for various PDE problems. Therefore, further investigation is required to understand the relationship between the PINN architectures and their performances. Moreover, there are a number of important

NeurIPS 2023 AI for Science Workshop.

hyperparameters for training PINNs, such as the learning rate, the number of training epochs, and the choices of optimizers. Manually tuning the architecture and hyperparameters is tedious and laborious. Therefore, we are motivated to study the following research question: *Can we automate the process of architecture and hyperparameters selection to improve the performance of PINNs?*

Despite the recent progress of automated hyperparameter tuning [1, 3] and neural architecture search (NAS) [13, 19, 5], automating the neural architecture design of PINNs remains an open and challenging problem. First, the search space that includes both discrete and continuous hyperparameters is extremely large. Existing hyperparameter optimization methods usually search the whole hyperparameter space, which can be inefficient. Second, the search objective for PINNs is unclear. Unlike other tasks that can naturally use the performance metric (e.g., accuracy) as the search objective, many PDEs may have no exact solutions such that the error values are not available. Therefore, we have to identify an alternative search objective.

To this end, we first conduct a comprehensive set of benchmarking pre-experiments to understand the search space by studying the relationship between each hyperparameter and the performance. We make two key observations from the experiments. *First*, we find that some design choices play a dominant role in the performance. For example, there is often a dominant activation function working better for each PDE. This motivates us to reduce the search space by decoupling it in a certain order. For instance, we can determine the best activation function with a small number of search trials, and then fix the activation function and focus on the search of the other hyperparameters. We observe similar phenomenons for other hyperparameters such as the changing point, depth, and width, which enables us to decouple them step by step. *Second*, we discover that the loss values are highly correlated with the errors. This makes the loss value a desirable search objective since it can be naturally obtained during the search for all the PDEs.

Based on the above observations, we propose Auto-PINN, the first automated machine learning framework to optimize the neural architecture and the hyperparameters of PINNs. Auto-PINN adopts a step-by-step decoupling strategy for search. Specifically, we search one hyperparameter at each step with the others fixed to one or a few sets of options. This strategy decreases the scale of the search space drastically. We perform extensive experiments to evaluate Auto-PINN on seven PDE benchmarks with different training data sampling methods. The quantitative comparison results show that Auto-PINN outperforms the other optimization strategies on both accuracy and stability with fewer search trials.

We summarize our main contributions as follows:

- We conduct a comprehensive set of benchmarking pre-experiments on the hyperparameter-performance relationships of PINNs. Our observations suggest we can significantly reduce the search space by decoupling the search of different hyperparameters. We also identify the loss value of PINNs as a desirable search objective.
- We propose Auto-PINN, the first automated neural architecture and hyperparameter optimization approach for PINNs. The decoupling strategy can substantially decrease the search complexity.
- We evaluate Auto-PINN on a series of PDE benchmarks and compare it with other baseline methods. The results suggest that Auto-PINN can consistently search PINN architectures that display good accuracy in different PDE problems, which outperforms other search algorithms,

## 2 Preliminaries

### 2.1 Benchmarking PDEs.

We select a set of standard PDE benchmarks for the experiments. We conducted pre-experiments on four representative PDEs. They are two diffusion (heat) equations, a wave equation and a Burgers' equation which are commonly used in PINN research [15]. We will refer to them as **Heat_0**, **Heat_1**, **Burgers**, and **Wave** in the following sections for conciseness. These PDEs include different kinds of differential operators and boundary/initial conditions, which are capable of illustrating regular rules for PINNs in the pre-experiments. Moreover, we design different data sampling schemes for each PDE to improve the credibility of the results. These PDEs are also involved in the formal comparison experiments, along with another three PDEs, which are two advection equations (**Advection_0** and **Advection_1**) and a reaction equation (**Reaction**). The details of these PDEs and the data sampling methods are shown in the Appendix A.

## 2.2 Search Space

In this section, we define the search space for the PINN architectures. Here we consider the following variables.

- *Width* and *Depth*. For an MLP-structured PINN, the *depth* is the number of hidden layers, and the *width* means the number of neurons in each hidden layer. We set the *width* ranging in $[16, 512]$ with the step of 8 (only for **Heat_0**) or in $[8, 256]$ with the step of 4 (for other PDEs). The *depth* ranges in $[3, 10]$ with the step of 1.
- *Activation Function*. The *activation functions* in PINNs determine different non-linear elements in the networks. We provide four options: Tanh, Sigmoid, ReLU, and Swish [21]. We leave the definitions of these activation functions in Appendix B.
- *Changing Point*. According to [15], PINNs can reach their best performance by training with an Adam optimizer in the first stage to get closer to the minimum and then switching to an L-BFGS second-order optimizer [12]. We need to decide the timing of that change. Therefore, we introduce a hyperparameter named *Changing Point* as a float number ranging from 0 to 1. This *changing point* indicates the proportion of the epochs using Adam to the total training epochs. For example, if the training epoch number is set to 10000 with a 0.4 *Changing Point*, that means the PINN will be trained with the Adam optimizer for $10000 \times 0.4 = 4000$ epochs, followed by 6000-epoch L-BFGS training. However, it makes little sense to search on a precise grid, so we only consider five discrete options $\{0.1, 0.2, 0.3, 0.4, 0.5\}$.

Initially, we do not include *learning rates* and the *training epochs* in the search. However, we will show results on those two training hyperparameters in Section 5.3, which indicate that they have a small effect on the final architecture search results.

## 3 Pre-Experiments and Observations

As we mentioned previously, neural architecture optimization for PINNs is still an under-explored problem. Therefore, we should first explore general rules for the hyperparameters of PINNs. Different from other deep learning tasks, the training strategy and the physical constraints of PINNs are unique. Therefore, we do not simply apply a hyperparameter search algorithm, but first do pre-experiments to understand the behavior of PINNs. For all experiments in this section, the PINNs are trained with 10000 training epochs, and the learning rate is set to $1e^{-5}$.

We first study the relationship between structure and performance of the PINNs. Throughout this paper, the main figure of merit to measure accuracy is the relative $L_2$ error. The reported errors are averages over three separate random initializations of the neural network weights in each case.

A set of heatmaps is shown in Figure 1 to display the L2 error results. More heatmap results are shown in Appendix C. We obtained several observations from these heatmaps:

**Observation 1** There is a dominant *activation function* in PINNs working better for each PDE, which can be easily found by searching a small subset of the whole space. For example, it is easy to see that Tanh is the best choice for **Heat_0**. Median error value across the subsets is a good metric to determine the dominant *activation function*.

**Observation 2** Under the dominant *activation function*, the larger *changing points* perform better or comparably than smaller values, as can be seen in the average error values shown beside the y-axes.

**Observation 3** The "wider and deeper PINNs are better" rule does not apply to all PDEs, e.g., the **Wave** PDE.

**Observation 4** The error distances (the values in parentheses in the cells) are usually very small, i.e., when the loss functions reach the smallest values in the training processes, so do the L2 error values.

Observations 1 and 2 suggest that it is possible to decouple the *activation function* and *changing point* from the search space. Observation 3 indicates that further research on the MLP structure is needed to determine *widths* and *depths* for PINNs. Observation 4 means that overfitting is not a problem for PINNs. The error value at the minimum loss function value is close to the minimum error that a PINN can actually reach. However, it is just a local summary for each cell. We need to compare the

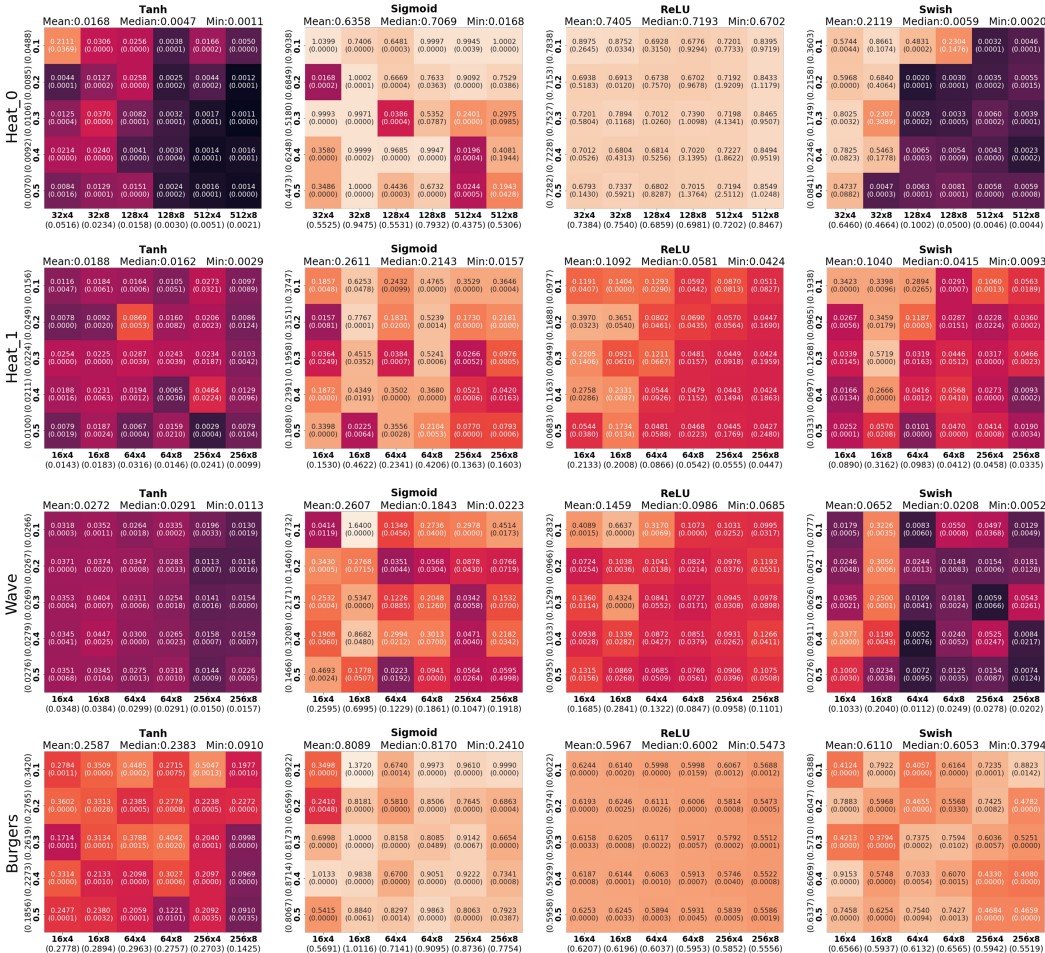

Figure 1: L2 Error heatmaps on different PDEs with different PINN configurations. Each row corresponds to a different PDE, and each column corresponds to a different activation function. The x-axis of each heatmap represents different *width* and *depth* settings of PINNs from small to larger scales. For instance, "$32 \times 4$" means the *width* is 32 and the *depth* is 4. The y-axis labels are different *changing points* from 0.1 to 0.5. For each cell in the heatmaps, the top number is the smallest L2 error value that the PINN actually reached in its training process. For a direct visual representation, the cells with deeper colors correspond to smaller error values, i.e., the PINNs have better performances. The number in the parentheses in each cell is the absolute distance from the smallest actual error value to the error value when the smallest loss function value is reached. On the top of each heatmap, we report the average, median, and minimum error values across the heatmap. The numbers in the parentheses around the x and y labels are mean error values across columns and rows, which show the average performance when the other hyperparameters are fixed.

loss-error relationship between cells to establish that the loss function is a good search objective over the entire space.

## 3.1 Structure-Error Relationship

*Width* and *depth* are key structure hyperparameters of PINNs. However, as mentioned in Observation 3, the pre-experiments above cannot give a clear relationship between the structures and error values of PINNs because of the low sampling rate in the spaces of the two hyperparameters. For that reason, we continue by doing more specific pre-experiments on the structure-error relationship.

The results are shown in Figure 2. Each data point (joined by lines) is the average L2 error over three random initializations. We fix the *activation function* and the *changing point* in each case and then sample the space of *width* and *depth*. More results can be found in Appendix C. Clearly, there are many *width* regions in which the best-performing network is not the deepest, e.g., *width* within $[400, 500]$ for the **Heat_0** and around 100 for the **Wave**. This confirms again Observation 3 that the "wider and deeper PINNs are better" rule does not always apply and that it is important to determine the optimal depth for a given width.

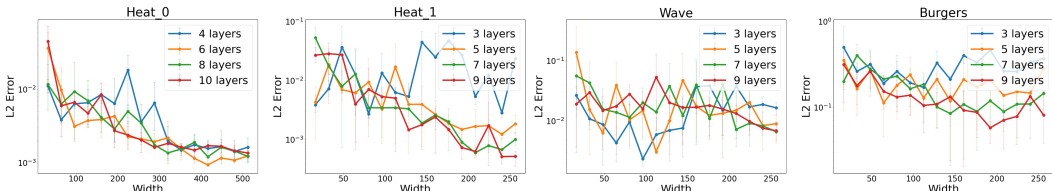

Figure 2: Structure-error relationship.

**Observation 5** Good regions in the space of *width* can be identified, and then one can search for the optimal *depth*.

### 3.2 Loss-Error Relationship

We would like to investigate if the loss function is a representative search objective across the entire configuration space. Hence, we report the loss-error relationship for different PDEs in Figure 3. The $x$-axis is the lowest loss value that a PINN reached in the training process, and the $y$-axis is the corresponding L2 error value. Note that Observation 4 has informed us that the smallest training error values agree with the smallest L2 error values. Each point in the figure represents a PINN architecture configuration shown in Figure 1. The R-square value indicates a strong linear correlation between the logarithmic loss and error values across the whole search space. Therefore, it is appropriate to leverage these loss values to judge the real performance of the PINNs. We have more linear correlation evidence shown in Appendix C. Hence, we have another observation:

**Observation 6** There is a strong linear correlation between the smallest log-loss values and the corresponding log-error values for the PINNs with different hyperparameter configurations in a PDE problem. We can take advantage of the training loss function values to assess the performance of the PINNs.

## 4 Auto-PINN: Automated Architecture Optimization for PINNs

The pre-experiments have provided us with a clear guideline on how to decouple the hyperparameter space, and we now can present our Auto-PINN approach. With the help of Observations 1–6, we are ready to decouple the hyperparameters in the large search space and find the best architectures with only a small number of search trials.

### 4.1 Methodology

**Input:** A PDE problem with training and testing data points. The hyperparameter search space is mentioned in Section 2.2.

**Search Objective:** The smallest loss values reached by the PINNs in the training processes (Observations 3 and 6).

**Step 0.** Set *changing points* to 0.5 (Observation 2) for the following **Step 1** to **Step 2.2**.

**Step 1.** Search the *activation function* (Observation 1). Sample on the *width* space exponentially and the *depth* space uniformly. Use the median search objective to determine the dominant *activation function*. This *activation function* will become the only choice in the following steps.

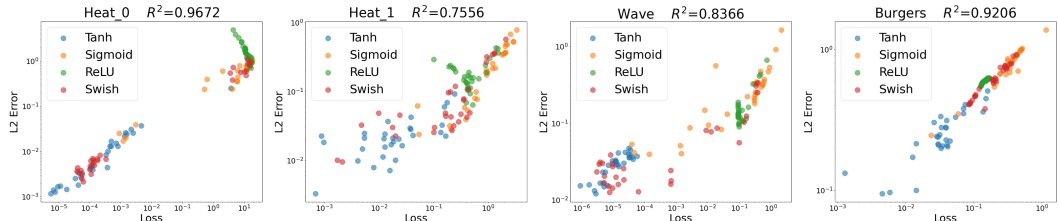

Figure 3: Loss-error relationship.

**Step 2.** Search the *depth* and *width*

    **Step 2.1.** Search the good-performance regions of *width* (Observation 5). Split the *width* space into several intervals. Sample several *width* settings in each interval to represent the performance of their interval, then combine them with the uniformly sampled *depth* settings. Use the median search objective to find several best *width* intervals as the good-performance regions.

    **Step 2.2.** Search the best *depth* settings. Collect all *width* settings inside the "good performance" regions with all *depth* settings. Search and find the top $K$ candidate structure configurations.

**Step 3.** Search the best *changing point* for each candidate.

**Step 4.** Verify the performance of the searched PINN architectures. Retrain every selected PINN five times with different initializations and report its median performance.

**Output:** $K$ different PINN architectures with small training loss values correlating with small testing errors (Observation 4).

Please refer to Appendix C to see the default settings of the Auto-PINN algorithm.

### 4.2 Complexity Analysis

The scale of the search space is greatly decreased by utilizing the proposed Auto-PINN algorithm. The entire search space in Section 2.2 contains $10,080$ possible configurations. Auto-PINN under the default settings only needs at most $261$ trials to find the best architectures, which is only $2.59\%$ to the whole search space. Detailed analysis is in Appendix C.

## 5 Experiments

### 5.1 Experimental Settings

In this section, we present the results of experiments that validate the effectiveness of the proposed method.

**PDE Benchmarks.** We conducted experiments using the **Heat_0**, **Heat_1**, **Wave**, **Burgers**, **Advection_0**, **Advection_1** and **Reaction** PDEs, which are described in Appendix A.

**Baseline Methods.** Random Search and HyperOpt [3] are selected as the baseline methods for comparison with Auto-PINN. We set 300 sampling numbers for the two baseline models, which are higher than the Auto-PINN.

**Implementation Details.** We set the learning rates to $1e-5$ and the number of training epochs to 10000. We implemented Auto-PINN and the baseline methods with the PINN package DeepXDE [15] and hyperparameter tuning package Tune [11]. DeepXDE [15] is a user-friendly open-sourced library for physics-informed machine learning including common PINNs and different training strategies. We use DeepXDE with some modifications in the APIs. Tune [11] is another Python library for experiment execution and hyperparameter tuning. It supports all mainstream machine learning frameworks and a large number of hyperparameter optimization methods. We utilized the trial parallelism feature of Tune to make our Auto-PINN more efficient. Underlying MLP models are built and trained with the PyTorch framework. We ran the experiments on 4 Nvidia 3090 GPUs.

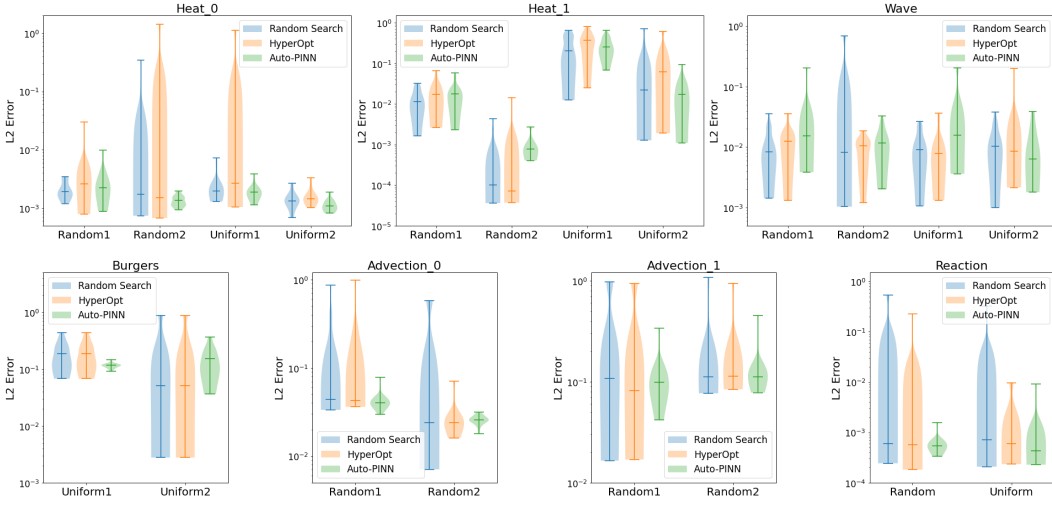

Figure 4: Comparison Results. The violin plots illustrate the range of L2 errors of the searched architectures in the PDE problems with different data sampling settings.

## 5.2 Comparison Results

The results are shown in Figure 4. Compared with the two baseline methods, the results show that Auto-PINN is more stable, as can be seen from the concentrated distributions. However, the Random Search and HyperOpt suffer from very large performance variances frequently. Meanwhile, Auto-PINN is capable of finding architectures with good performance. In most cases, the architectures with the median error values found by Auto-PINN match or exceed the performances of the baseline methods with fewer search trials. In summary, Auto-PINN outperforms the baseline methods in both stability and accuracy with fewer search trials.

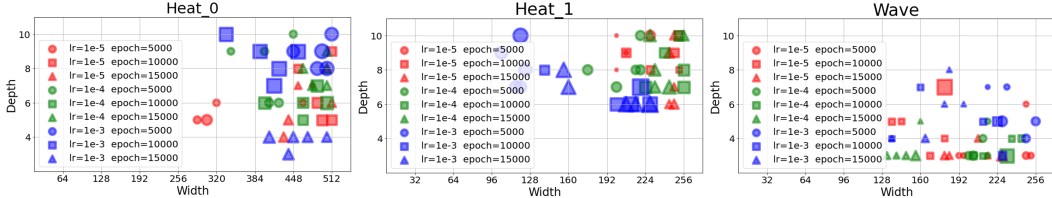

Figure 5: Searched architecture distributions by Auto-PINN with different learning rates and epoch settings. The sizes of the markers display the relative performances of each PINN configuration. Larger markers mean better performance, that is, the corresponding PINNs have smaller testing error values.

## 5.3 Influence of Learning Rates and Training Epochs

As we mentioned, we do not consider the learning rates and training epochs in our search space. Further experimental results indicate that Auto-PINN is not sensitive to these two training hyperparameters. As shown in Figure 5, the searched architectures with different learning rates and epochs congregate at a specific region in the search space, which means those architectures searched by Auto-PINN are still available within a range of proper learning rates and epochs. Therefore, there is no need to search again with different learning rates and the numbers of epochs. On the other hand, we can see that the PDEs show different preferences in the structures. For example, the **Heat_0** PDE requires wider structures but is insensitive to the *depth*, whereas the **Wave** PDE is not sensitive to *width* but prefers *shallower* PINNs. Auto-PINN is able to identify consistent architectures for different PDEs, which is a very important point for future research.

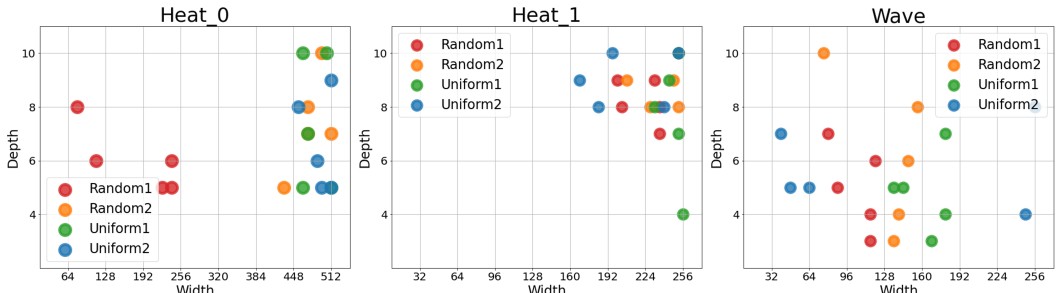

Figure 6: Searched architecture distributions by Auto-PINN with different data sampling methods.

### 5.4 Influence of Data Sampling

Different PDEs have distinct sensitivity to data sampling, as illustrated in Figure 6, where we employed random and uniform sampling schemes, each using two different sampling densities, for the collocation, boundary ,and initial points; see Appendix A for the details. The results confirm that the structure-sampling relationship is distinct for each PDE. Therefore, it is not appropriate to simply reuse the searched architectures when the sampling strategy changes. We remark that more sophisticated adaptive- and residual-based data sampling methods [7, 17, 18] for PINNs have been proposed. These sampling methods can be used as an internal setting for Auto-PINN to achieve better search accuracy in future work.

## 6 Conclusion and Future Work

In this paper, we proposed Auto-PINN, the first systematic neural architecture and hyperparameter optimization approach for PINNs, which can search for the best architectures and hyperparameters for different PDE problems within a large search space. We conducted a comprehensive set of pre-experiments to understand the search space of PINNs. Based on the observations, we proposed a step-by-step decoupling strategy to reduce the search space and use the loss value as the search objective. The comparison results demonstrate the stability and the effectiveness of Auto-PINN. In addition, we perform experiments to analyze the influences of the learning rate, the training epochs, and the data sampling strategies. We found that the performance is not sensitive to the learning rates and the training epochs, while the best configuration depends on the adopted sampling strategy. We hope the insights gleaned from our observations can motivate future exploration in PINNs and that Auto-PINN can serve as a strong baseline in future research. In future work, we plan to incorporate more sophisticated data sampling strategies into the search space of Auto-PINN to achieve better performances.

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

# Appendix A    Benchmarking PDEs and Data Samplings

In this appendix, we present the PDEs and the data sampling strategies used in our experiments.

## A.1    Benchmarking PDEs

We utilize some standard PDE benchmarks in our experiments, including two heat equations, a wave equation, a Burgers' equation, two advection equations, and a reaction equation.

**Heat_0:** The heat equation describes the heat or temperature distribution in a given domain over time. This heat equation contains Dirichlet boundary conditions.

Equation:
$$\frac{\partial u(x,t)}{\partial t} - \frac{\partial^2 u(x,t)}{\partial x^2} = -e^{-t}(\sin \pi x - \pi^2 \sin \pi x), \; x \in [-1,1], \; t \in [0,1] \tag{1}$$

Boundary Condition: $\qquad\qquad\qquad u(-1,t) = u(1,t) = 0 \tag{2}$

Initial Condition: $\qquad\qquad\qquad u(x,0) = \sin \pi x \tag{3}$

Solution: $\qquad\qquad\qquad u(x,t) = e^{-t} \sin \pi x \tag{4}$

**Heat_1:** This heat equation contains Neumann boundary conditions.

Equation:
$$\frac{\partial u(x,t)}{\partial t} - \frac{\partial^2 u(x,t)}{\partial x^2} = 1 + x \cos t, \; x \in [0,1], \; t \in [0,1] \tag{5}$$

Boundary Condition:
$$\frac{\partial u(x,t)}{\partial x}\Big|_{x=0} = \frac{\partial u(x,t)}{\partial x}\Big|_{x=1} = \sin t \tag{6}$$

Initial Condition: $\qquad\qquad\qquad u(x,0) = 1 + \cos 2\pi x \tag{7}$

Solution: $\qquad\qquad u(x,t) = 1 + t + e^{-4\pi^2 t} \cos 2\pi x + x \sin t \tag{8}$

**Wave:** The wave equation describes the propagation of oscillations in a space, such as mechanical and electromagnetic waves. This wave equation contains both Dirichlet and Neumann conditions.

Equation:
$$\frac{\partial^2 u(x,t)}{\partial t^2} - \frac{\partial^2 u(x,t)}{\partial x^2} = x \cos t, \; x \in [0,1], \; t \in [0,1] \tag{9}$$

Boundary Condition:
$$u(0,t) = \sin t, \; \frac{\partial u(x,t)}{\partial x}\Big|_{x=0} = \cos t - \sin t + t \tag{10}$$

Initial Condition:
$$u(x,0) = \sin x, \; \frac{\partial u(x,t)}{\partial t}\Big|_{t=0} = \cos x \tag{11}$$

Solution: $\qquad\qquad u(x,t) = \sin(x+t) + x(t - \sin t) \tag{12}$

**Burgers:** Burgers equation has been leveraged to model shock flows, wave propagation in combustion chambers, vehicular traffic movement, and more. We use an accurate approximation of the solution[4].

Equation:
$$\frac{\partial u(x,t)}{\partial t} + u(x,t)\frac{\partial u(x,t)}{\partial x} - \frac{0.01}{\pi}\frac{\partial^2 u(x,t)}{\partial x^2} = 0, \; x \in [-1,1], \; t \in [0, \frac{5}{\pi}] \tag{13}$$

Boundary Condition: $\qquad\qquad\qquad u(-1,t) = u(1,t) = 0 \tag{14}$

Initial Condition: $\qquad\qquad\qquad u(x,0) = -\sin \pi x \tag{15}$

**Advection_0:** The advection equation describes the motion of a scalar field as it is advected by a known velocity vector field.

Equation:
$$\frac{\partial u(x,t)}{\partial t} + \frac{\partial u(x,t)}{\partial x} = 0, \; x \in [0,2], \; t \in [0,1] \tag{16}$$

Boundary Condition: $\qquad\qquad u(0,t) = 2, \; u(2,t) = 0 \tag{17}$

Initial Condition:
$$u(x,0) = \begin{cases} 2, & 0 \le x < 1 \\ 0, & 1 < x \le 2 \end{cases} \tag{18}$$

Solution:
$$u(x,t) = \begin{cases} 2, & 0 \le x < 1+t \\ 0, & 1+t < x \le 2 \end{cases} \tag{19}$$

Table 1: Data sampling methods for each PDE. "Random" means pseudo-random sampling and "Uniform" is sampling on a uniform grid in the spatio-temporal domain. The numbers after each sampling method name distinguish between different sampling densities.

| PDEs | Sampling Methods | # Collocation Points | # Boundary Points | # Initial Points |
|------|------------------|----------------------|--------------------|------------------|
| **Heat_0** | Random1 | 105 | 40 | 20 |
|  | Random2 | 512 | 200 | 100 |
|  | Uniform1 | 105 | 40 | 20 |
|  | Uniform2 | 512 | 200 | 100 |
| **Heat_1** | Random1 | 400 | 100 | 50 |
|  | Random2 | 2500 | 500 | 250 |
|  | Uniform1 | 400 | 100 | 50 |
|  | Uniform2 | 2500 | 500 | 250 |
| **Wave** | Random1 | 2025 | 200 | 200 |
|  | Random2 | 5041 | 500 | 500 |
|  | Uniform1 | 2025 | 200 | 200 |
|  | Uniform2 | 5041 | 500 | 500 |
| **Burgers** | Uniform1 | 5040 | 500 | 500 |
|  | Uniform2 | 10057 | 1000 | 1000 |
| **Advection_0** | Random1 | 200 | 40 | 20 |
|  | Random2 | 800 | 160 | 80 |
| **Advection_1** | Random1 | 200 | 40 | 20 |
|  | Random2 | 800 | 160 | 80 |
| **Reaction** | Random | 800 | 160 | 80 |
|  | Uniform | 800 | 160 | 80 |

**Advection_1:** This advection equation has different boundary conditions than **Advection_0**.

Equation:
$$\frac{\partial u(x,t)}{\partial t} + \frac{\partial u(x,t)}{\partial x} = 0,\ x \in [0,2],\ t \in [0,1] \tag{20}$$

Boundary Condition:
$$u(0,t) = 1,\ u(2,t) = -1 \tag{21}$$

Initial Condition:
$$u(x,0) = \begin{cases} 1, & 0 \le x < 1 \\ -1, & 1 < x \le 2 \end{cases} \tag{22}$$

Solution:
$$u(x,t) = \begin{cases} 1, & 0 \le x < 1+t \\ -1, & 1+t < x \le 2 \end{cases} \tag{23}$$

**Reaction:** The reaction equation describes chemical reactions.

Equation:
$$\frac{\partial u(x,t)}{\partial t} = u(x,t)(1 - u(x,t)),\ x \in [0,2],\ t \in [0,1] \tag{24}$$

Boundary Condition:
$$u(0,t) = \frac{e^{-1}e^{t}}{e^{-1}e^{t} + 1 - e^{-1}} \tag{25}$$

Initial Condition:
$$u(x,0) = e^{-(x-1)^2} \tag{26}$$

Solution:
$$u(x,t) = \frac{e^{-(x-1)^2}e^{t}}{e^{-(x-1)^2}e^{t} + 1 - e^{-(x-1)^2}} \tag{27}$$

## A.2 Data Sampling Methods

In our experiments, we consider different sampling methods including random samplings and uniform samplings. We provide the details of the different data sampling methods in Table 1.

# Appendix B  Activation Functions

In this appendix, we present the mathematical expressions of the four *activation functions* in the search space.

**Tanh:**

$$\tanh(x) = \frac{e^x - e^{-x}}{e^x + e^{-x}} \tag{28}$$

**Sigmoid:**

$$\text{sigmoid}(x) = \frac{1}{1 + e^{-x}} \tag{29}$$

**ReLU:**

$$\text{relu}(x) = \max(0, x) \tag{30}$$

**Swish[21]:**

$$\text{swish}(x) = x \cdot \text{sigmoid}(x) \tag{31}$$

# Appendix C   Additional Pre-Experiment Results and More Details about Auto-PINN

## C.1   Additional Pre-Experiment Results

In this section, we provide additional results to further support the observations in Section 3.

**Additional Error Heatmap Results.** We report additional error heatmap results in Figure 7, 8, 9 and 10 for different PDEs. The heatmaps in rows follows the order of the sampling methods in Table 1. The heatmaps shown in the main paper are also included for reference.

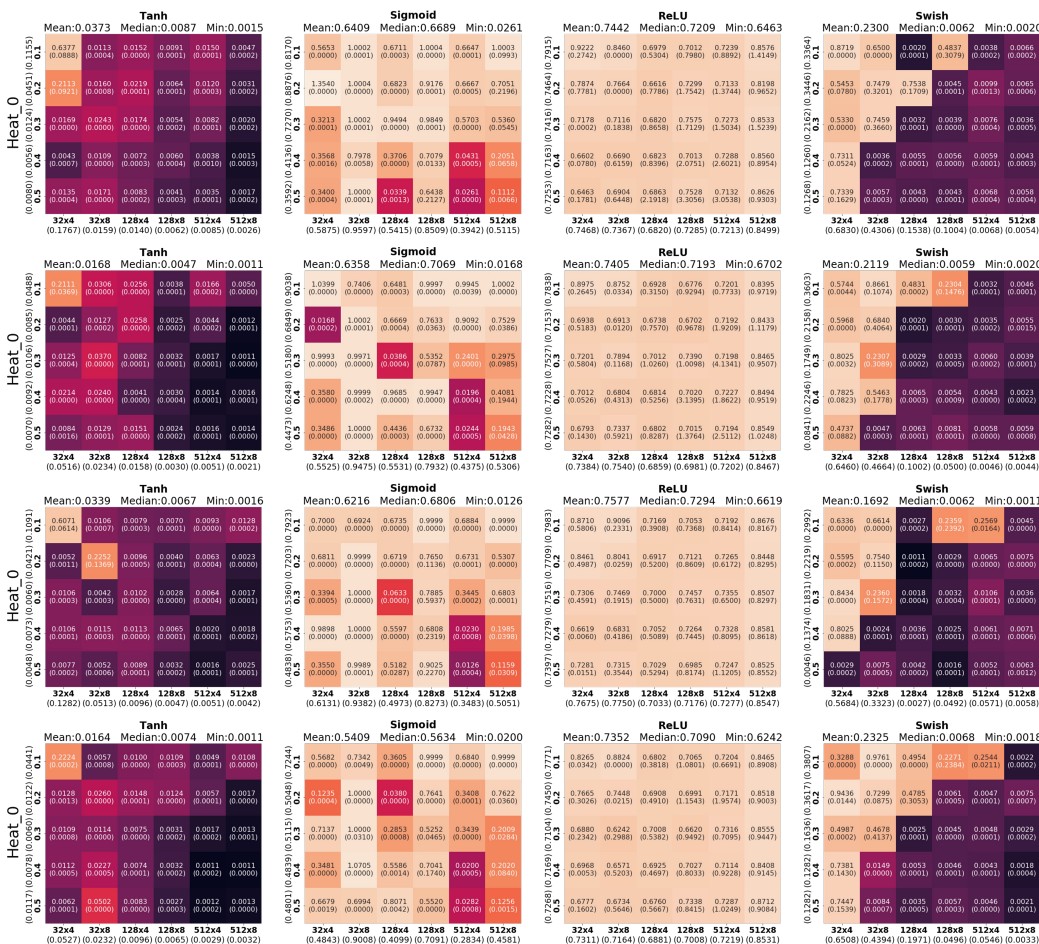

Figure 7: Error heatmaps for **Heat_0**.

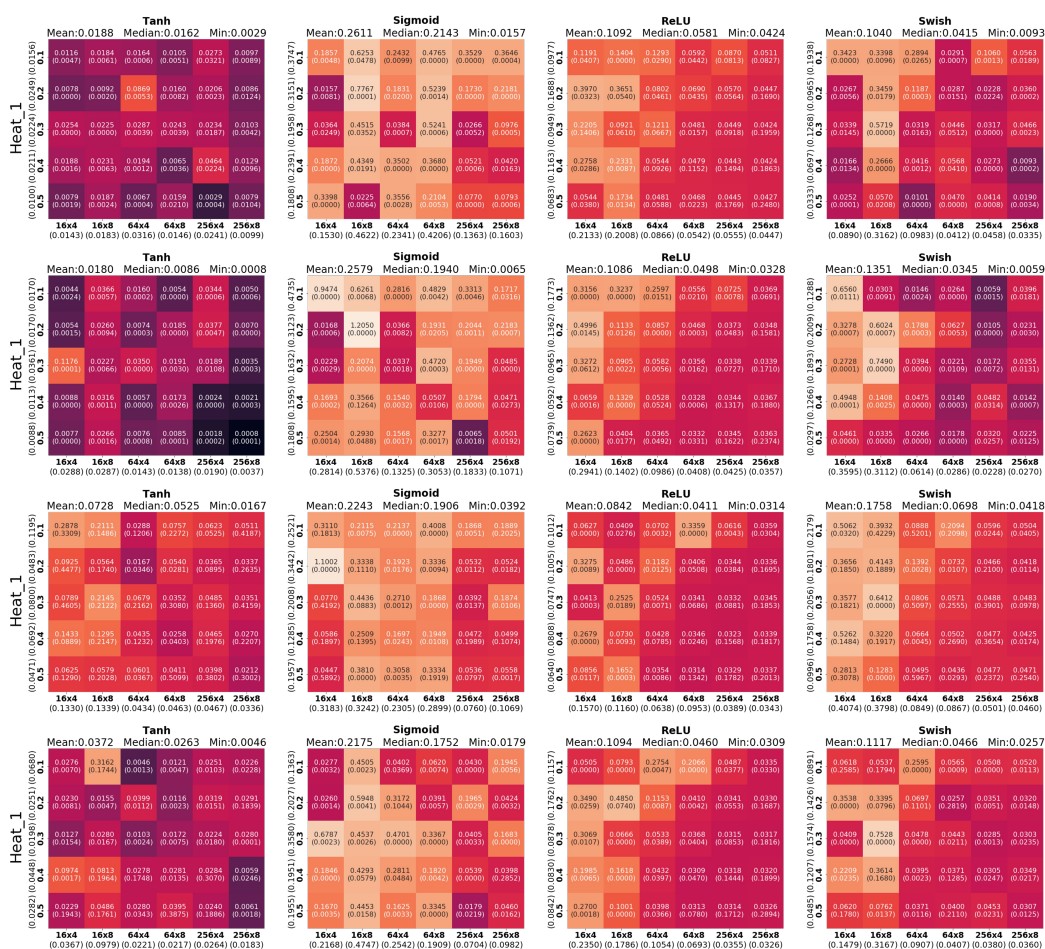

Figure 8: Error heatmaps for **Heat_1**.

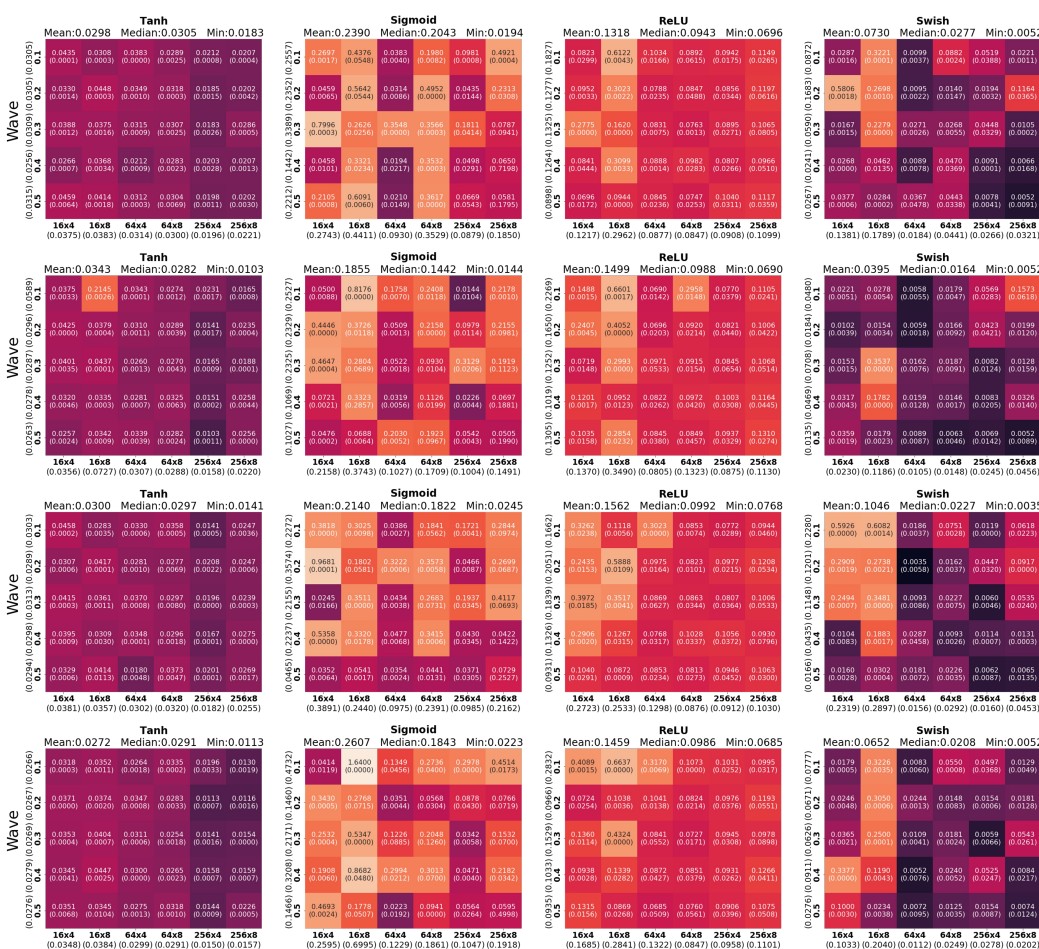

Figure 9: Error heatmaps for **Wave**.

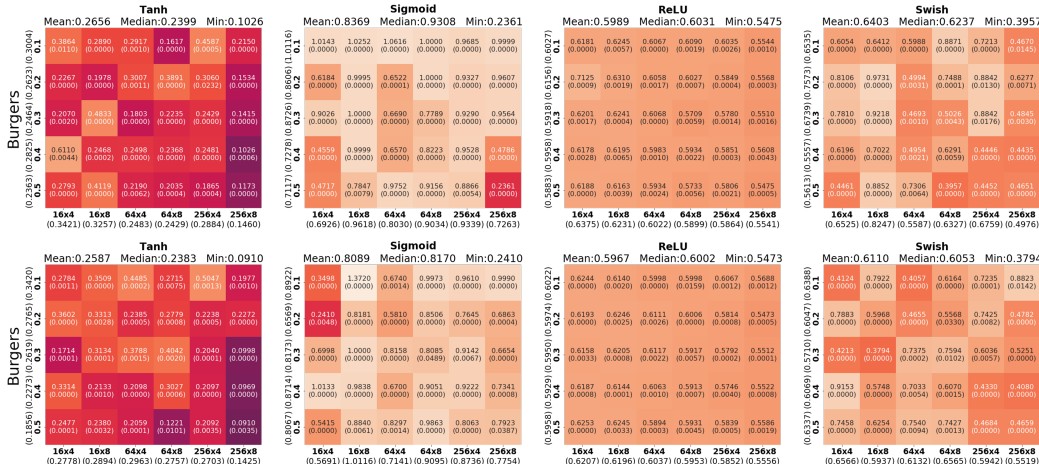

Figure 10: Error heatmaps for **Burgers**.

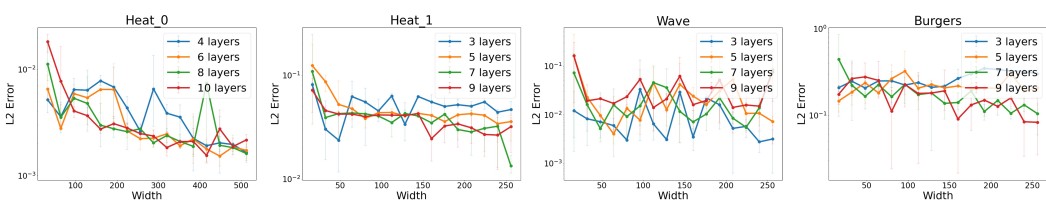

Figure 11: Structure-error relationship.

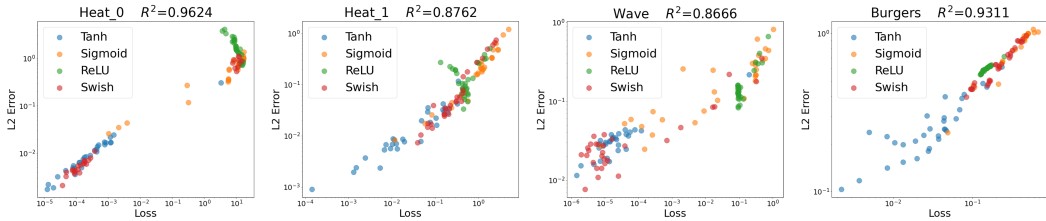

Figure 12: Loss-error relationship.

**Additional Structure-Error Relationship Results.** We report additional structure-error relationship results in Figure 11.

**Additional Loss-Error Relationship Results.** We report additional loss-error relationship results in Figure 12.

## C.2    More Details about Auto-PINN

In this section, we present more details about the proposed method, Auto-PINN.

**Default Settings.** Here we present the default setting of our proposed method.

**Step 0.** Set *changing points* to **0.5** (Observation 2) for the following **Step 1** to **Step 2.2**.

**Step 1.** Search the *activation function* (Observation 1). Sample **3** *widths* exponentially and **3** *depths* uniformly. Use the median search objective to determine the dominant *activation function*. This *activation function* will become the only choice in the following steps.

**Step 2.** Search the *depth* and *width*.

**Step 2.1.** Search the good-performance regions of *width* (Observation 5). Split the *width* space into **8** intervals uniformly. Sample **3** *width* settings in each interval randomly to represent the performance of their interval, then combine them with the uniformly sampled **3** *depth* settings. Use the median search objective to find the top **2** *width* intervals as the good-performance regions.

**Step 2.2.** Search the best *depth* settings. Collect all $8 + 8 = 16$ or $8 + 7 = 15$ *width* settings inside the top **2** "good performance" regions with all **8** *depth* settings. Search and find the top **5** candidate structure configurations.

**Step 3.** Search the best *changing point* for each candidate from the **5** choices.

**Step 4.** Verify the performance of the searched PINN architectures. Retrain every selected PINN five times with different intializations and report its median performance.

**Complexity Analysis.** Here we provide a detailed complexity analysis of the number in search trials.

In the default search space, we provide **4** *activation functions*, **5** options of the *changing points*, **8** *widths* and **63** *depths*. Therefore, the entire search space contains $4 \times 5 \times 8 \times 63 = 10,080$ configurations.

Under the default Auto-PINN settings, without considering the **Step 4** for verification, the pipeline needs $3 \times 3 \times 4 = 36$ trials in **Step 1**, $3 \times 8 \times 3 = 72$ trials in **Step 2.1**, at most $(8 + 8) \times 8 = 128$ trials in **Step 2.2** and $5 \times 5 = 25$ in **Step 3**. Therefore, Auto-PINN requires at most $36 + 72 + 128 + 25 = 261$ search trials in total, which is $261/10,080 = 2.59\%$ of the whole search space.

**Best Architectures Searched by Auto-PINN.** Here we give a list to show the architecture searching results by Auto-PINN. The diversity of the searched best architectures can be shown in Table 2, which suggests the effectiveness of Auto-PINN. All the experiment settings are the same as those in the main paper. We set the learning rates to $1e-5$ and the number of training epochs to $10000$.

Table 2: Best architectures searched by Auto-PINN for each PDE and sampling method, according to smallest L2 Error among the top $5$ results.

| PDEs | Sampling Methods | *Width* | *Depth* | *Activation Function* | *Changing Point* |
|------|------------------|---------|---------|----------------------|------------------|
| **Heat_0** | Random1 | 80 | 8 | Swish | 0.5 |
| | Random2 | 512 | 7 | Tanh | 0.4 |
| | Uniform1 | 464 | 5 | Tanh | 0.5 |
| | Uniform2 | 496 | 5 | Tanh | 0.4 |
| **Heat_1** | Random1 | 236 | 8 | Tanh | 0.5 |
| | Random2 | 248 | 9 | Tanh | 0.5 |
| | Uniform1 | 232 | 8 | Tanh | 0.5 |
| | Uniform2 | 252 | 10 | Tanh | 0.4 |
| **Wave** | Random1 | 116 | 4 | Swish | 0.4 |
| | Random2 | 180 | 7 | Tanh | 0.4 |
| | Uniform1 | 136 | 3 | Swish | 0.5 |
| | Uniform2 | 40 | 7 | Tanh | 0.4 |
| **Burgers** | Uniform1 | 256 | 10 | Tanh | 0.5 |
| | Uniform2 | 212 | 10 | Tanh | 0.4 |
| **Advection_0** | Random1 | 48 | 7 | ReLU | 0.5 |
| | Random2 | 16 | 5 | Tanh | 0.2 |
| **Advection_1** | Random1 | 256 | 4 | Tanh | 0.4 |
| | Random2 | 84 | 5 | Tanh | 0.1 |
| **Reaction** | Random | 32 | 3 | Tanh | 0.4 |
| | Uniform | 156 | 4 | Swish | 0.2 |

