# OpenReview forum: "Auto-PINN: Understanding and Optimizing Physics-Informed Neural Architecture"
_NeurIPS.cc/2023/Workshop/AI4Science — NeurIPS2023-AI4Science Poster_

### Official Review · Reviewer_t6uV · 2023-10-07
**Very nice paper with a clear review and writing, highlighting on the systematic design of parameter selection in PINNs**

**Rating:** 8
**Confidence:** 5

**Review:**

The authors discussed the choice of width/ depth, activation function, and changing point of optimizer when training with PINN. They provided a systematic review along with some observations on the parameters. Those observations in turn reduce complexity when searching for optimal parameters.

Extensive numerical experiments on the benchmarks were conducted, and a clear methodology was proposed. The complexity of the auto-selection is analyzed. Overall it is very pertinent and very well written.

Maybe one small criticism of the paper is its novelty. It seems to the referee that those observations and tuning processes are more or less known within the community. However, such paper is very helpful to the community

---

### Official Review · Reviewer_tA2G · 2023-10-25

**Rating:** 5
**Confidence:** 3

**Review:**

Summary

This work introduces Auto-PINN, a hyperparameter optimization procedure curated for Physics-Informed Neural Networks (PINNs) solvers for partial differential equations (PDE). The search space for hyperparameters is lowered by running pre-experiments and by serially optimizing hyperparameters over a shorter range. The experiment is conducted across different flavors of PDEs and the results show that for certain hyperparameters (like network width and network depth) the optimum value is dependent on the PDE type.

Comments:

1. The work tackles a prevalent issue of hyperparameter tuning in a more systematic fashion.

2. A key step before the Auto-PINN is implemented is the pre-experiments that are conducted. If I understand correctly, any new flavor of PDE which is used would warrant a pre-experiment step to identify a subset of relevant activation functions, network architecture dimensions etc. This is non-trivial and does not facilitate adoption of Auto-PINN readily for more real life scenarios without going though the pre-experiment step.
3. In the pre-experiment step itself, a handful of options are considered within the search space (pre-defined set of activation functions, width and depth ranges etc). These are selected based on prior experience with PINN PDE solvers, I assume. In case of a new problem statement with a fairly unique PDE, what is the best way of selecting the hyperparameter search space at the pre-experiment step?


While the work has merit and has shown a systematic way of reducing the search space, it does rely on a non-trivial pre-experiment step which needs to be executed for any new flavor of PDE. The Auto-PINN itself is more like a curated list of steps to perform a systematic search instead of a stand-alone tool which can be easily incorporated for hyperparameter tuning tasks.

---

### Meta-Review · Area_Chair_osqS · 2023-10-26

**Recommendation:** Accept (Poster)
**Confidence:** 4

**Metareview:**

In this paper, authors have claimed to propose the first systematic, automated hyperparameter optimization approach for Physics-Informed Neural Networks (PINNs), called as Auto-PINN, which employs Neural Architecture Search techniques for PINN design. Auto-PINN is proposed based on observations from benchmarking experiments on standard PDE benchmarks. The paper discusses an important problem, and is clearly written and organised.
Key concerns are about need of pre-experiments and limited hyperparameter considerations, however, I think the paper is a good step in direction of understanding and optimising PINNs so I recommend acceptance.